# Importance of sex and gender factors for COVID-19 infection and hospitalisation: a sex-stratified analysis using machine learning in UK Biobank data

Zahra Azizi,[1] Yumika Shiba,[2] Pouria Alipour,[1,3] Farhad Maleki,[4] Valeria Raparelli,[5,6] Colleen Norris,[6,7] Reza Forghani,[4] Louise Pilote  ,[1,3,8] Khaled El Emam  ,[9,10] The GOING-FWD investigators

LP and KEE are joint senior authors.

For numbered affiliations see end of article.

**Correspondence to**
Dr Louise Pilote;
louise.pilote@mcgill.ca and
Dr Khaled El Emam;
kelemam@ehealthinformation.ca

## ABSTRACT

**Objective** To examine sex and gender roles in COVID-19 test positivity and hospitalisation in sex-stratified predictive models using machine learning.

**Design** Cross-sectional study.

**Setting** UK Biobank prospective cohort.

**Participants** Participants tested between 16 March 2020 and 18 May 2020 were analysed.

**Main outcome measures** The endpoints of the study were COVID-19 test positivity and hospitalisation. Forty-two individuals' demographics, psychosocial factors and comorbidities were used as likely determinants of outcomes. Gradient boosting machine was used for building prediction models.

**Results** Of 4510 individuals tested (51.2% female, mean age=68.5±8.9 years), 29.4% tested positive. Males were more likely to be positive than females (31.6% vs 27.3%, p=0.001). In females, living in more deprived areas, lower income, increased low-density lipoprotein (LDL) to high-density lipoprotein (HDL) ratio, working night shifts and living with a greater number of family members were associated with a higher likelihood of COVID-19 positive test. While in males, greater body mass index and LDL to HDL ratio were the factors associated with a positive test. Older age and adverse cardiometabolic characteristics were the most prominent variables associated with hospitalisation of test-positive patients in both overall and sex-stratified models.

**Conclusion** High-risk jobs, crowded living arrangements and living in deprived areas were associated with increased COVID-19 infection in females, while high-risk cardiometabolic characteristics were more influential in males. Gender-related factors have a greater impact on females; hence, they should be considered in identifying priority groups for COVID-19 infection vaccination campaigns.

## INTRODUCTION

The novel coronavirus SARS-CoV-2 (COVID-19) pandemic has led to more than 250 million reported positive cases and over 5 million deaths worldwide as of November 2021.[1] As vaccination is rolling out, continuous

## STRENGTHS AND LIMITATIONS OF THIS STUDY

⇒ A unique feature of the study is the investigation of numerous psycho-socio-cultural factors in conjunction with clinical and laboratory factors made feasible through machine learning algorithms.

⇒ The assessment of sex-stratified algorithms to elucidate the most influential factors in both sexes adds novel information as it has rarely been done to date.

⇒ The first limitation of this study is selection bias due to the lack of systematic and random testing across the UK.

⇒ This analysis was done using the baseline diagnostic data that were collected from 2006 to 2010; therefore, misclassification of determinants is possible. However, previous studies of the UK Biobank have shown a high correlation between baseline and follow-up data for a subsample of patients who had further visits for imaging.

⇒ Finally, the relatively low predictive model performance (test positivity: area under curve (AUC): 0.570 (95% CI: 0.537 to 0.604), hospitalisation: AUC: 0.60 (95% CI: 0.534 to 0.665))—as expected—reflects other influences not captured in the UK Biobank.

efforts are made to establish risk factors for the disease and find vulnerable populations across the globe.[2–8]

There has been an imbalance in infection susceptibility, severity, and mortality between males and females.[9] These differences are multifactorial and have been attributed to a combination of biological (ie, genetic, hormonal) and psycho-socio-cultural differences (gender).[9–13] While 'sex' refers to a set of biological attributes in humans and animals, 'gender' refers to the roles, behaviours and identities of individuals that form throughout life.[14 15] It is increasingly recognised that both sex and gender play significant roles in health outcomes,[14 15] including the acquisition of infections and response to infection and

treatments.[16] The WHO statement,[16] addressing sex and gender in epidemic-prone infectious diseases, outlines that differences between males and females can lead to differences in activity patterns in work and in family roles, which may increase the risk of exposure to infectious disease in a particular setting.[16] Therefore, lack of consideration of the influences of sex and gender in COVID-19 contraction can potentially hinder the effectiveness of COVID-19 vaccination prioritisation strategies.

With the introduction of the COVID-19 vaccines, identifying those most vulnerable to infection with accurate prediction models that account for more complex relationships is urgent. Machine learning algorithms can explore non-linear and complex relationships by considering the interaction between both biological and psycho-socio-cultural factors together; however, these methods are still underused in the medical field, and few predictive models have been tailored to each sex.[17–19] Therefore, we examined sex-related and gender-related factors associated with SARS-CoV-2 test positivity and COVID-19 hospitalisation in the UK Biobank (UKB) cohort and developed sex-stratified predictive models using machine learning methods.

## METHODS
### Study population
This is a cross-sectional study of UKB data. The UKB (https://www.ukbiobank.ac.uk/) is a prospective cohort study that collects health, lifestyle, genetic and imaging data for over 500 000 randomly selected participants in the UK.[20] Baseline data collection took place between 2006 and 2010 across England, Scotland and Wales. The age of participants at recruitment was between 40 and 69 years.[20] Data were collected in 22 assessment centres through four main methods: touchscreen questionnaires, verbal interviews, physical measures and biological sampling.[20] For this study, only data from England between 16 March 2020 and 18 May 2020 were used due to the unavailability of COVID-19 test results from Scotland and Wales at the time of analysis.

### Patient and public involvement
No patient involved in the design, or conduct, or reporting, or dissemination plans of our research.

### COVID-19 test results
COVID-19 test results were available from Public Health England data for 4510 participants, which was linked to UKB baseline data.[21] The primary endpoint of the study was test positivity. We defined testing positive as having at least one positive test result. The test results were available from 16 March 2020 to 18 May 2020. The secondary endpoint of the study was being hospitalised for a COVID-19-related illness. For this purpose, we chose test-positive patients who had at least one positive result in an inpatient setting. The results are from samples taken from the combined nose, throat, sputum or lower respiratory tract.

The analysis for infection was done using real-time PCR tests for SARS-CoV-2.

### Baseline characteristics
Patients' demographics, psychosocial (gender-related variables), anthropometric variables and comorbidities that were collected at baseline 2006–2010 were used for analysis. We used this particular baseline data for socio-economic status (SES) and occupation since studies have shown a high correlation between baseline and follow-up data for a subsample of patients who had further visits for imaging.[22]

### Gender-related psychosocial variables
A multistep approach for identifying gender-related variables was exploited based on GOING-FWD (Gender Outcomes INternational Group: to Further Well-being Development) methodology,[23] the Women Health Research Network's gender framework (ie, gender identity, gender roles, gender relationships and institutionalised gender), and the Canadian Institutes of Health Research sex and gender modules.[14 15 24] Based on this approach and data availability, the following variables were selected: multiple deprivation indices (online supplemental file 1), employment, jobs involving night shift, education level, smoking status, alcohol consumption, number of children, household crowding, housing ownership, income, leisure and social activities, risk-taking behaviours and neuroticism score. Details of the variables are available in the UKB data dictionary (https://biobank.ctsu.ox.ac.uk/crystal/search.cgi).

### Comorbidities
Baseline comorbidities were self-reported and collected using a touch screen device—that is, hypertension, diabetes, chronic obstructive pulmonary disease, asthma, allergy, history of stroke, heart attack, angina, deep vein thrombosis, pulmonary embolism and cancer. In addition, the number of medications and long-standing illness, disability, or infirmity were also included and coded as dichotomous variables. These variables were selected based on their significance as demonstrated by a number of previous investigations on the UKB.[2 3 5–8 18 22 25–27]

### Physical and biological characteristics
Variables used as measures of physical and biological characteristics were body mass index (BMI) (defined as weight $(kg)/height (m)^2$), waist to hip ratio (WHR), levels of vitamin D, haemoglobin A1c (HbA1c), high-density lipoprotein (HDL) and low-density lipoprotein (LDL); all coded as continuous variables.

### Statistical analyses
Descriptive statistics were reported as mean and SD for continuous variables and frequency and percentage for categorical variables. Group-based differences (test negative vs positive) in baseline characteristics were compared using independent Student's t-test for continuous variables (or its non-parametric counterpart for skewed

**Table 1** Descriptive characteristics of COVID-19 test-positive versus test-negative males and females

| Baseline characteristics | Overall | | | | Females | | | Males | | |
|---|---|---|---|---|---|---|---|---|---|---|
| | All | Test − | Test + | P value | Test − | Test + | P value | Test − | Test + | P value |
| | N=4510 | N=3184 | N=1326 | | N=1679 | N=630 | | N=1505 | N=696 | |
| **Demographics** | | | | | | | | | | |
| Age group, % | | | | | | | | | | |
| <60 years | 23.6 | 21.8 | 28 | <0.001* | 26.4 | 34.1 | 0.001* | 16.5 | 22.4 | 0.006 |
| ≥60–<70 | 24.7 | 25.4 | 23 | | 26.6 | 25.6 | | 24 | 20.7 | |
| ≥70–<80 | 45.9 | 47.1 | 42.8 | | 42 | 35.1 | | 52.9 | 49.9 | |
| ≥80 | 5.9 | 5.7 | 6.2 | | 4.9 | 5.2 | | 6.6 | 7 | |
| Sex, % | | | | | | | | | | |
| Female | 51.2 | 52.7 | 47.5 | 0.001* | | | | | | |
| Ethnicity, % | | | | | | | | | | |
| White | 90.7 | 92.4 | 86.8 | <0.001* | 91.5 | 86.4 | <0.001* | 93.4 | 87.1 | <0.001* |
| Mixed | 2.3 | 2.2 | 2.4 | | 2.6 | 2.2 | | 1.7 | 2.6 | |
| Asian | 3.3 | 2.6 | 5 | | 2.1 | 5.3 | | 3.1 | 4.8 | |
| Black British | 3.7 | 2.9 | 5.8 | | 3.8 | 6.1 | | 1.9 | 5.5 | |
| Country of birth, % | | | | | | | | | | |
| UK | 88.9 | 90.2 | 85.7 | <0.001* | 89.1 | 84 | 0.001* | 91.5 | 87.4 | 0.003 |
| **Biological variables** | | | | | | | | | | |
| Number of medications | 3.3±3.3 | 3.3±3.3 | 3.2±3.3 | 0.7 | 3.2±3.2 | 3.2±3.3 | 0.8 | 3.3±3.4 | 3.3±3.3 | 0.6 |
| BMI (kg/m$^2$) | 28.4±5.4 | 28.2±5.4 | 28.9±5.4 | <0.001* | 28.1±5.8 | 28.7±6.2 | 0.02 | 28.3±4.9 | 29.1±4.7 | <0.001* |
| WHR | 0.9±0.1 | 0.8±0.09 | 0.8±0.09 | <0.001* | 0.8±0.07 | 0.8±0.07 | 0.03 | 0.9 | 0.9 | 0.1 |
| HTN, % | 35.3 | 34.5 | 37 | 0.1 | 28.8 | 31.7 | 0.1 | 41 | 41.7 | 0.7 |
| Stroke, % | 3 | 3 | 3 | 1 | 1.7 | 2.1 | 0.6 | 4.4 | 3.8 | 0.5 |
| Heart attack, % | 4 | 4.1 | 3.8 | 0.7 | 1.4 | 1.8 | 0.6 | 7.1 | 5.7 | 0.2 |
| Angina, % | 5.7 | 5.9 | 5.3 | 0.4 | 3.3 | 3.4 | 1 | 8.8 | 7.1 | 0.2 |
| DVT, % | 3.3 | 3.1 | 4 | 0.1 | 2.9 | 4.1 | 0.1 | 3.3 | 3.8 | 0.6 |
| PE, % | 1.7 | 1.9 | 1.1 | 0.09 | 1.9 | 1.4 | 0.6 | 1.9 | 0.9 | 0.09 |
| COPD, % | 3.2 | 3.2 | 3.1 | 0.9 | 2.6 | 2.6 | 1 | 3.9 | 3.6 | 0.8 |
| Asthma, % | 13.5 | 13.6 | 13 | 0.5 | 16.4 | 16.3 | 0.9 | 10.5 | 10 | 0.7 |
| Allergy, % | 21.9 | 22.8 | 19.8 | 0.03 | 26.2 | 22.6 | 0.08 | 19 | 17.3 | 0.3 |
| Diabetes, % | 9.8 | 9.6 | 10.3 | 0.5 | 7.4 | 8 | 0.7 | 12 | 12.4 | 0.8 |
| Cancer, % | 9.4 | 10.1 | 7.7 | 0.01 | 11.1 | 9.2 | 0.2 | 9.1 | 6.4 | 0.4 |

Continued

**Table 1** Continued

| Baseline characteristics | Overall | | | | Females | | | Males | | |
|---|---|---|---|---|---|---|---|---|---|---|
| | All | Test – | Test + | P value | Test – | Test + | P value | Test – | Test + | P value |
| | N=4510 | N=3184 | N=1326 | | N=1679 | N=630 | | N=1505 | N=696 | |
| Disability, % | 45.6 | 46.1 | 44.3 | 0.2 | 42.2 | 38.9 | 0.1 | 50.3 | 49.2 | 0.6 |
| Vitamin D level (nmol/L) | 46.5±21.2 | 46.6±21 | 46.1±21.6 | 0.4 | 45.7±20.7 | 45.2±20.9 | 0.6 | 47.6±21.4 | 46.7±22.2 | 0.4 |
| HbA1c level (mmol/mol) | 37.3±8.5 | 37.1±8.3 | 37.8±9.1 | 0.03 | 36.6±7.7 | 36.7±8.1 | 0.7 | 37.7±8.9 | 38.7±9.8 | 0.02 |
| LDL level (mmol/L) | 3.4±0.8 | 3.4±0.8 | 3.4±0.8 | 0.7 | 3.4±0.8 | 3.5±0.8 | 0.5 | 3.3±0.8 | 3.3±0.8 | 0.8 |
| HDL level (mmol/L) | 1.4±0.3 | 1.4±0.3 | 1.3±0.3 | <0.001* | 1.5±0.3 | 1.4±0.3 | <0.001* | 1.3±0.3 | 1.2±0.2 | 0.004 |
| LDL to HDL ratio | 2.6±0.9 | 2.5±0.9 | 2.7±0.8 | <0.001* | 2.3±0.8 | 2.5±0.8 | <0.001* | 2.8±0.9 | 2.8±0.8 | 0.3 |

Results are presented as mean±SD for numerical variables and percentage for categorical variables.
*Bonferroni correction=0.05/24 (number of variables)=0.002; based on this correction, the differences marked with * are considered statistically significant.
BMI, body mass index; COPD, chronic obstructive pulmonary disease; DVT, deep vein thrombosis; HbA1c, haemoglobin A1c; HDL, high-density lipoprotein; HTN, hypertension; LDL, low-density lipoprotein; PE, pulmonary embolism; WHR, waist to hip ratio.

distributions) and $X^2$ for categorical variables. P values of less than or equal to 0.05 were considered statistically significant. A complete case analysis (pairwise deletion) approach was used for dealing with missing data in the descriptive analysis. Bonferroni type adjustments were used to correct for multiple testing and the results after adjustment were used for interpretation. Data analysis was performed using R-Studio (V.1.3.1093) and R software (V.4.0.3).

## Machine learning

The data were split into 70% training and 30% test sets, where the test set was only used for the evaluation of the final models. The training set was used to develop gradient boosting decision tree models[28–30] using the 'gbm3' R package for predicting SARS-CoV-2 test positivity and hospitalisation prediction. To reduce the effect of class imbalance on model development,[31] a bootstrap oversampling approach was used. For each endpoint, that is, test positivity or hospitalisation, three different models were developed: (1) a model for males, (2) a model for females, and (3) an overall model for males and females combined. Calibration was then performed using isotonic regression method.

Using k-fold cross-validation is considered a best practice in developing machine learning models.[32] Therefore, for developing each model, a 10-fold cross-validation was performed on the training set. We also used a grid search procedure to find the best combination hyperparameters (eg, learning rate, interaction depth, bagging fraction and the minimum number of observations in terminal nodes) (online supplemental file 1) using fivefold cross-validation on the training dataset and the area under the receiver operating characteristic curve (AUROC) metric as the criterion. A Bernoulli distribution was used for classification models. These resulted in three trained models for SARS-CoV-2 test positivity and three models for SARS-CoV-2 hospitalisation.

To provide an unbiased estimate of the model generalisation errors, the performance of the trained models was assessed and reported on the test set. Confusion matrix-derived metrics including accuracy, precision, recall (sensitivity), specificity as well as area under curve (AUC) score were used as the performance measures.[33] We used the best threshold of receiver operating characteristic curve (ROC) as cut-off for determining precision/recall and sensitivity/specificity. Another metric which focuses on predictions of the positive class is the Area Under the Precision-Recall Curve (AUPRC).[34] Interpretation of AUPRC is dependent on the class distribution of the outcome as the minimal achievable value is dependent on that distribution,[35] and the AUPRC value of a random classifier is the rate of the positive class.[35]

## Most influential variables

For identifying the most influential variables, permutation-based feature importance was used.[36] This approach measures a feature (variable) importance by

**Table 2** Gender variables in patients who tested positive compared with test-negative patients in overall, males and females

| Gender variables | Overall | | | | Female | | | Male | | |
|---|---|---|---|---|---|---|---|---|---|---|
| | All N=4474 | Test – N=3161 | Test + N=1313 | P value | Test – N=1668 | Test + N=625 | P value | Test – N=1493 | Test + N=688 | P value |
| **Behavioural/lifestyle risk factors** | | | | | | | | | | |
| Smoking, % | | | | | | | | | | |
| 0=never | 48.5 | 48.3 | 49 | 0.02 | 54.5 | 58.7 | 0.1 | 41.3 | 40.1 | <0.001* |
| 1=previous | 38.4 | 37.8 | 40 | | 34.5 | 30.9 | | 41.5 | 48.3 | |
| 2=current | 13.1 | 14 | 11 | | 11 | 10.4 | | 17.2 | 11.6 | |
| Alcohol, % | | | | | | | | | | |
| 0=never | 11.6 | 10.9 | 13.3 | <0.001* | 11.8 | 16.4 | 0.001* | 9.9 | 10.6 | 0.002* |
| 1=occasionally | 14.1 | 14.1 | 14.2 | | 18.4 | 18.3 | | 9.3 | 10.4 | |
| 2=1–3 times/month | 11.8 | 11 | 13.9 | | 14 | 16.9 | | 7.7 | 11.1 | |
| 3=1–2/week | 24.1 | 23.6 | 25.4 | | 24.1 | 23.8 | | 23.1 | 26.8 | |
| 4=3–4/week | 19 | 19.6 | 17.4 | | 16.2 | 14.1 | | 23.4 | 20.4 | |
| 5=daily/almost daily | 19.4 | 20.8 | 15.8 | | 15.6 | 10.5 | | 26.7 | 20.7 | |
| **Gender-related variables** | | | | | | | | | | |
| Employment status, % | | | | | | | | | | |
| Employed | 50.4 | 49.6 | 52.4 | 0.09 | 53.7 | 58.6 | 0.04 | 45 | 46.7 | 0.5 |
| Night shift, % | | | | | | | | | | |
| 0=never/rarely | 45.1 | 47.5 | 41.4 | 0.21 | 49 | 40.9 | 0.17 | 45.6 | 42 | 0.8 |
| 1=sometimes | 32.1 | 31.6 | 33 | | 31.5 | 32.5 | | 31.7 | 33.6 | |
| 2=usually/always | 22.8 | 20.9 | 25.6 | | 19.5 | 26.6 | | 22.8 | 24.4 | |
| Number of children | 1.9±1.4 | 1.8±1.4 | 1.9±1.4 | 0.03 | 1.8±1.2 | 1.9±1.2 | 0.15 | 1.8±1.5 | 1.9±1.6 | 0.12 |
| Education, % | | | | | | | | | | |
| None/NVQ/HND/HNC/CSEs | 36.4 | 34.4 | 41.3 | <0.001* | 31.2 | 37.5 | 0.02 | 38 | 44.8 | 0.02 |
| O-level/ GCSEs | 19.2 | 19.8 | 17.7 | <0.001* | 20.8 | 19.2 | | 18.6 | 16.3 | 0.001* |
| A-levels/AS-levels | 9.9 | 9.9 | 9.8 | <0.001* | 10.5 | 10.6 | | 9.2 | 9.1 | 0.01 |
| College/university/professional | 34.5 | 35.9 | 31.2 | <0.001* | 37.5 | 32.6 | | 34.2 | 29.8 | <0.001* |
| Deprivation indices | | | | | | | | | | |
| Education score | 18.6±17.9 | 17.7±17.4 | 20.6±19.3 | <0.001* | 17.6±17.1 | 20.9±19.3 | <0.001* | 17.9±17.6 | 20.4±18.9 | 0.003 |
| Environmental score | 20.6±16.4 | 20±15.7 | 22.1±17.8 | <0.001* | 19.9±15.6 | 21.3±17.9 | 0.08 | 20.1±15.9 | 22.7±17.6 | 0.001* |
| Employment score | 0.1±0.06 | 0.09±0.06 | 0.1±0.07 | <0.001* | 0.09±0.06 | 0.1±0.07 | <0.001* | 0.10±0.06 | 0.11±0.07 | 0.01 |
| Health score | 0.09±0.9 | 0.05±0.9 | 0.2±0.8 | <0.001* | 0.05±0.8 | 0.1±0.9 | 0.02 | 0.06±0.9 | 0.2±0.08 | <0.001* |
| Housing score | 19.7±10.2 | 19.8±10.3 | 19.7±9.9 | 0.7 | 19.8±10.2 | 19.9±9.9 | 0.7 | 19.8±10.5 | 19.4±9.9 | 0.4 |
| Income score | 0.1±0.11 | 0.1±0.1 | 0.1±0.11 | <0.001* | 0.1±0.1 | 0.1±0.12 | <0.001* | 0.1±0.11 | 0.1±0.11 | 0.008 |
| Income, % | | | | | | | | | | |

Continued

**Table 2** Continued

| Gender variables | Overall | | | | Female | | | Male | | |
|---|---|---|---|---|---|---|---|---|---|---|
| | All | Test − | Test + | P value | Test − | Test + | P value | Test − | Test + | P value |
| | N=4474 | N=3161 | N=1313 | | N=1668 | N=625 | | N=1493 | N=688 | |
| 1=≤18 000 | 32 | 31.5 | 33.5 | 0.1 | 31.2 | 32.1 | 0.17 | 31.8 | 34.7 | 0.26 |
| 2=18 000–30 900 | 24.6 | 24.5 | 24.6 | | 23.3 | 24 | | 25.8 | 25.2 | |
| 3=31 000–51 900 | 22.3 | 22.2 | 22.6 | | 23.2 | 26.5 | | 21.1 | 19.1 | |
| 4=52 000–100 000 | 16.5 | 16.7 | 16 | | 17.5 | 14.3 | | 16 | 17.5 | |
| 5=≥100 000 | 4.6 | 5.1 | 3.3 | | 4.8 | 3.2 | | 5.3 | 3.5 | |
| Number in household | 2.4±1.5 | 2.3±1.5 | 2.5±1.3 | 0.001* | 2.4±1.8 | 2.5±1.4 | 0.04 | 2.3±1.1 | 2.4±1.3 | 0.006 |
| House ownership, % | 81.4 | 82.2 | 79.6 | 0.04 | 84.5 | 80.4 | 0.02 | 79.6 | 78.8 | 0.7 |
| Neuroticism score | 4.3±3.3 | 4.3±3.3 | 4.2±3.3 | 0.4 | 4.8±3.3 | 4.8±3.2 | 0.9 | 3.8±3.2 | 1.7±3.3 | 0.5 |
| Risk-taking behaviour, % | 30.7 | 29.9 | 32.5 | 0.1 | 23.3 | 27 | 0.7 | 37.3 | 37.5 | 0.9 |
| Leisure/social activities, % | | | | | | | | | | |
| 0=no | 32.4 | 32.7 | 31.8 | 0.003 | 32.9 | 34.3 | 0.03 | 32.4 | 29.6 | 0.09 |
| 1=sports club or gym | 24.1 | 24.6 | 23 | | 25.8 | 21.9 | | 23.2 | 24.1 | |
| 2=pub or social club | 21.1 | 20.4 | 22.6 | | 13.6 | 16.1 | | 28 | 28.6 | |
| 3=religious group | 10.4 | 9.5 | 12.5 | | 11.5 | 14.8 | | 7.3 | 10.4 | |
| 4=adult education class | 2.8 | 3.1 | 2.2 | | 4.4 | 3.7 | | 1.5 | 0.9 | |
| 5=other group activities | 9.1 | 9.7 | 7.8 | | 11.8 | 9.3 | | 7.5 | 6.4 | |

Results are presented as mean±SD for numerical variables and percentage for categorical variables.
*Bonferroni correction=0.05/18 (number of variables)=0.003; based on this correction, the differences marked with * are considered statistically significant.
CSE, Certificate of Secondary Education; GCSE, General Certificate of Secondary Education; HNC, Higher National Certificat; HND, Higher National Diploma; NVQ, National Vocational Qualification.

**Table 3** Summary of GBM model performance for predicting test-positive results in training (10-fold cross-validation) and test sets

| | | Accuracy | AUC* | Threshold† | PRAUC | Precision | Recall | Specificity |
|---|---|---|---|---|---|---|---|---|
| Overall | Test set | 0.56 | 0.570 (0.537 to 0.604) | 0.25 | 0.33 | 0.34 | 0.56 | 0.56 |
| Male | Test set | 0.53 | 0.575 (0.529 to 0.621) | 0.32 | 0.38 | 0.37 | 0.67 | 0.46 |
| Female | Test set | 0.62 | 0.561 (0.512 to 0.609) | 0.26 | 0.31 | 0.32 | 0.35 | 0.72 |

*AUC is reported with 95% CI.
†Used the best threshold of ROC as cut-off for determining precision/recall, sensitivity/specificity.
AUC, area under curve; GBM, gradient boosting machine; PRAUC, Area Under the Precision-Recall Curve .

calculating the increase of the model's prediction error after permuting the feature.

We reported partial dependence plots (PDPs) using the 'pdp' package in R to understand the marginal effect of a feature on the predicted outcome. PDP demonstrates how the response variable changes as we change the value of a feature while taking into account the average effect of all the other features in the model.[37] The Y axis shows how the predicted value changes with change in predictor variables. If the line in the plot is constant near zero, it means that the variable has no effect on the model. A negative value means that a specific value of the predictor variable is less likely to predict the correct class of outcome, whereas a positive value means the predictor variable has a positive impact on predicting the correct class.[38] A locally estimated scatterplot smoothing line is fit to show the trend.

## RESULTS

Of 4510 patients (51.2% females, and 68.5±8.88 years) who were tested, 29.4% were positive. Females were less likely to be positive (males: 31.6% vs females: 27.3%, p=0.001). In descriptive analyses, there was a difference in age between test-positive and test-negative individuals (p<0.001); specifically, those younger than 60 years (test positive vs negative: 28% vs 21.8%) and those older than 80 years (test + vs −: 6.2% vs 5.7%). Similarly, there was significant difference in test positivity among ethnicities (minority ethnicity: test + vs −: 13.2% vs 7.6% p<0.001)

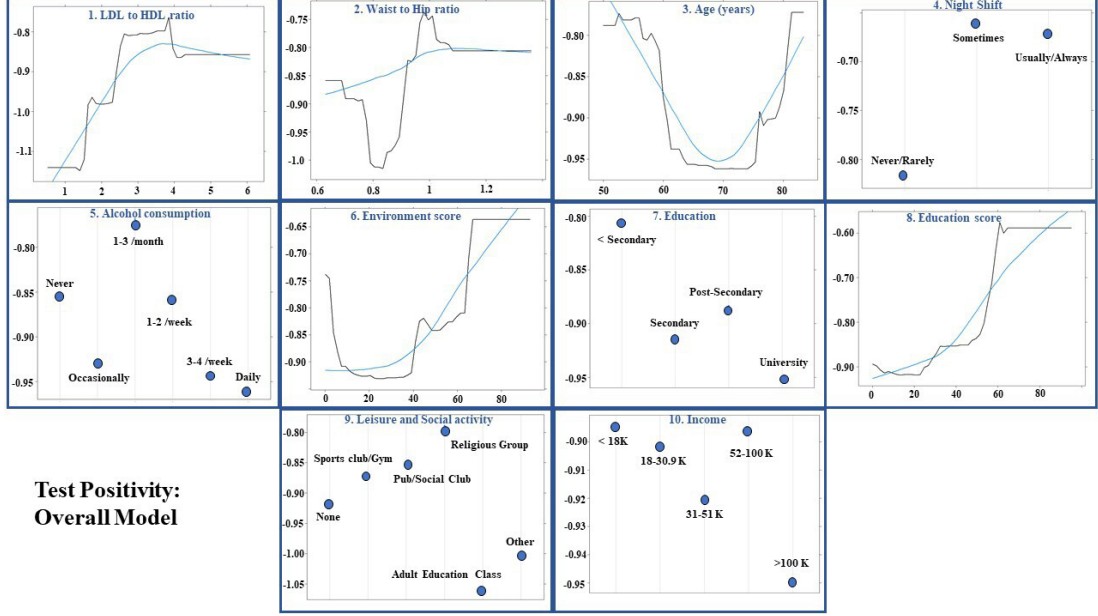

**Figure 1** Overall results: partial dependency plots for predicting test-positive results for first 10 most influential variables using permutation methods in the overall population using shrinkage=0.01, bag fraction=0.5, interaction depth=5, cross-validation fold=10, train fraction=0.7 and n.minobsinnode=10 as hyperparameters from grid search. The X axis is the predictor variable in the model. The Y axis shows how the predicted value changes with change in predictor variables. If the line in the plot is constant near zero, it means that the variable has no effect on the model. A negative value means that a specific value of the predictor variable is less likely to predict the correct class of outcome, whereas a positive value means the predictor variable has a positive impact on predicting the correct class. A locally estimated scatterplot smoothing line is fit to show the trend. The number beside each variable shows the order of feature importance and most influential variables for each model. For identifying the most influential variables, permutation-based feature importance was used. This approach measures a feature (variable) importance by calculating the increase of the model's prediction error after permuting the feature. HDL, high-density lipoprotein; LDL, low-density lipoprotein.

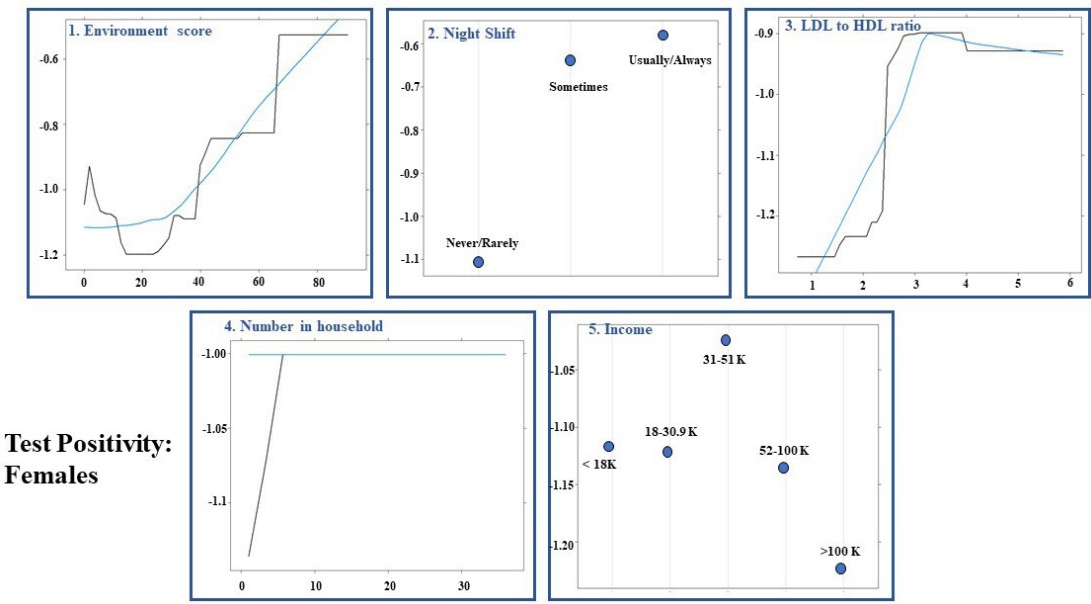

**Figure 2** Results for females: partial dependency plots for predicting test-positive results for first five most influential variables using permutation methods in females using shrinkage=0.01, bag fraction=1, interaction depth=7, cross-validation fold=10, train fraction=0.8 and n.minobsinnode=15 as hyperparameters from grid search. HDL, high-density lipoprotein; LDL, low-density lipoprotein.

and individuals born outside of the UK (test + vs −: 14.3% vs 9.8%, p<0.001). Moreover, infected patients had a higher BMI (test + vs −: 28.9±5.4 vs 28.2±5.4, p<0.001), WHR (test + vs −: 0.89±0.09 vs 0.88±0.09), HbA1c (test + vs −: 37.8±9.1 vs 37.1±8.3, p=0.03), LDL to HDL ratio (test + vs −: 2.7±0.8 vs 2.5±0.9, p<0.001) and lower HDL (test + vs −: 1.3±0.3 vs 1.4±0.3, p<0.001) levels. On the other hand, smokers (test + vs −: 11% vs 14%,p=0.02), alcohol drinkers (test + vs −: 15.8% vs 20.8%, p<0.001),

more educated individuals (test + vs −: 31.2% vs 35.9%, p<0.001), house owners (test + vs −: 79.6% vs 82.2%, p=0.04) and those who did not participate in any social activity (test + vs −: 31.8% vs 32.7%, p=0.003) had a lower rate of infection. Test-positive patients tended to have a greater number of people in their household (test + vs −: 2.5±1.3 vs 2.3±1.5, p<0.001) and live in more deprived areas (p<0.001). These results held true in a sex-stratified analysis (tables 1 and 2).

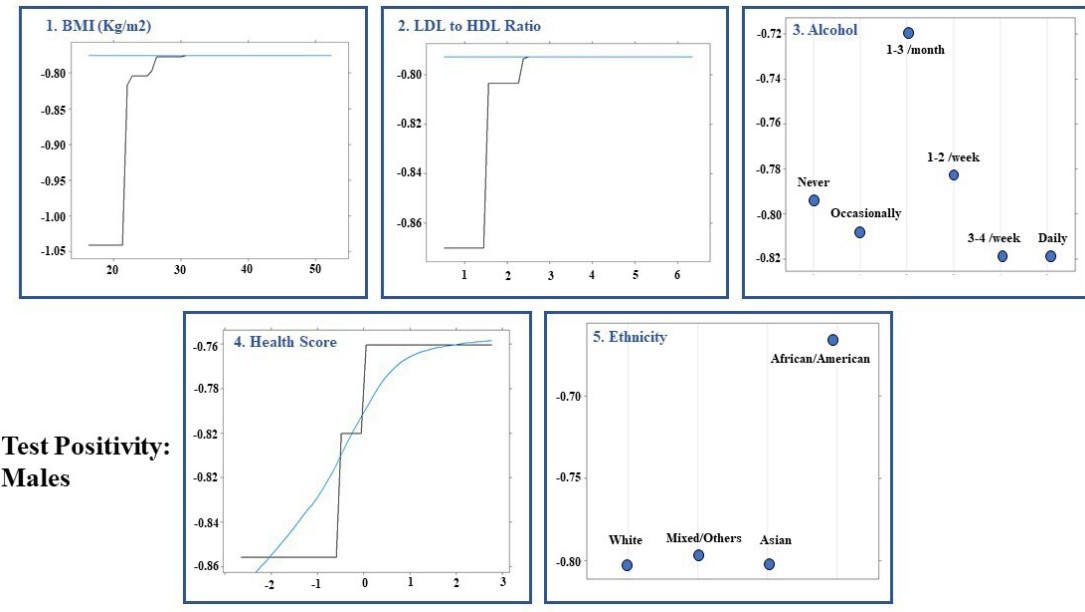

**Figure 3** Results for males: partial dependency plots for predicting test positive results for first five most influential variables using permutation methods in males using shrinkage=0.001, bag fraction=1, interaction depth=3, cross-validation fold=10, train fraction=0.8 and n.minobsinnode=5 as hyperparameters from grid search. BMI, body mass index; HDL, high-density lipoprotein; LDL, low-density lipoprotein.

**Table 4** Summary of GBM model performance for predicting being hospitalised in test-positive patients in training (10-fold cross-validation) and test sets

| | | Accuracy | AUC* | Threshold† | PRAUC | Precision | Recall | Specificity |
|---|---|---|---|---|---|---|---|---|
| Overall | Test set | 0.61 | 0.60 (0.534 to 0.665) | 0.69 | 0.75 | 0.77 | 0.66 | 0.48 |
| Male | Test set | 0.51 | 0.544 (0.453 to 0.635) | 0.75 | 0.76 | 0.85 | 0.43 | 0.75 |
| Female | Test set | 0.55 | 0.612 (0.532 to 0.692) | 0.68 | 0.75 | 0.81 | 0.47 | 0.75 |

*AUC is reported with 95% CI.
†Used the best threshold of ROC as cut-off for determining precision/recall, sensitivity/specificity.
AUC, area under curve; GBM, gradient boosting machine; PRAUC, Area Under the Precision-Recall Curve .

### Machine learning-based prediction models for SARS-CoV-2 positive test

The AUCs for test positivity in the overall model, male and female-specific models were 0.570 (95% CI: 0.537 to 0.604), 0.575 (95% CI: 0.529 to 0.621) and 0.561 (95% CI: 0.512 to 0.609), respectively. The performance of the gradient boosted decision tree models is summarised in table 3. Figures 1–3 illustrate the order of variable importance and partial dependence plots used for interpreting the results and direction of each variable in the models. The prediction models for the overall study population suggest that an increased LDL to HDL ratio, WHR and age were associated with a higher likelihood of test positivity. Additionally, individuals who worked night shifts or lived in a more deprived area (lower environment and education scores) as well as those participating in social activities—including sports clubs, bars and religious groups—had a higher risk of having a positive test. In contrast, individuals with higher education levels, higher income, and those with daily or almost daily alcohol consumption were less likely to have a positive result.

The sex-specific models for test positivity showed that gender factors were more important in females, whereas in males, biological factors were significant contributors to test positivity. Females who lived in more deprived areas (increased environment score), had increased LDL to HDL ratio, worked night shifts and had a greater number of family members in their household were more likely to test positive. Moreover, those with income greater than 100 000 were less likely to test positive (figure 2). In comparison, males with greater BMI and LDL to HDL ratio, more deprived area (greater score) and black British ethnicity were more likely to test positive (figure 3).

### Machine learning-based prediction models for COVID-19-related hospitalisation

The AUCs for hospitalisation in test-positive patients in overall, male, and female-specific models were 0.60 (95% CI: 0.534 to 0.665), 0.544 (95% CI: 0.453 to 0.635), and 0.612 (95% CI: 0.532 to 0.692), respectively. The performance of the gradient boosted decision tree models is summarised in table 4. Among the 1326 test-positive

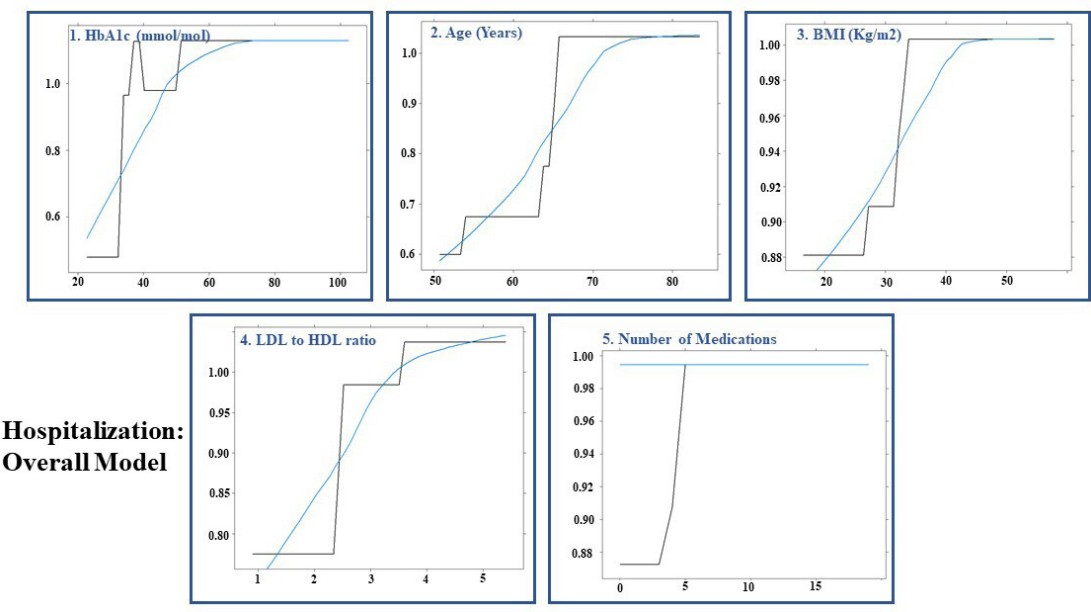

**Figure 4** Hospitalisation in overall population: partial dependency plots for predicting hospitalisation in test-positive patients for first five most influential variables using permutation methods using shrinkage=0.1, bag fraction=0.8, interaction depth=3, cross-validation fold=10, train fraction=0.7 and n.minobsinnode=15 as hyperparameters from grid search. BMI, body mass index; HbA1c, haemoglobin A1c; HDL, high-density lipoprotein; LDL, low-density lipoprotein.

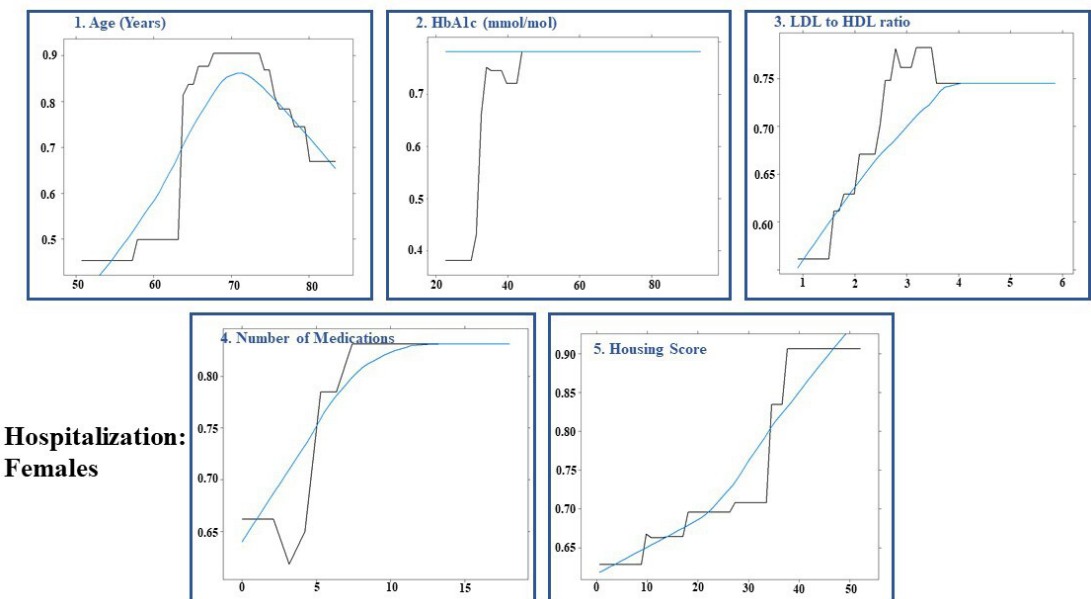

**Figure 5** Hospitalisation of females: partial dependency plot for predicting hospitalisation in test-positive females for first five most influential variables using permutation methods using shrinkage=0.05, bag fraction=0.65, interaction depth=7, cross-validation fold=10, train fraction=0.8 and n.minobsinnode=15 as hyperparameters from grid search. HbA1c, haemoglobin A1c; HDL, high-density lipoprotein; LDL, low-density lipoprotein.

patients, 932 (70.3%) were hospitalised (females: 413 (44.3%)). Figures 4–6 illustrate the order of variable importance and partial dependence plots used for interpreting the results and direction of each variable in the models. The result of the overall model to predict hospitalisation in test-positive patients revealed that those with higher HbA1c level, older age, greater BMI, higher LDL to HDL ratio and greater number of medications had greater risk of being hospitalised (figure 4).

The sex-stratified model revealed that older age, a higher level of HbA1c, LDL to HDL ratio, a greater number of medications and higher housing score (showing more deprived areas) were most influential variables in predicting hospitalisation in test-positive females (figure 5); whereas older age, an increased HbA1c level, WHR, LDL to HDL ratio and BMI were the most influential variables associated with hospitalisation in test-positive males (figure 6).

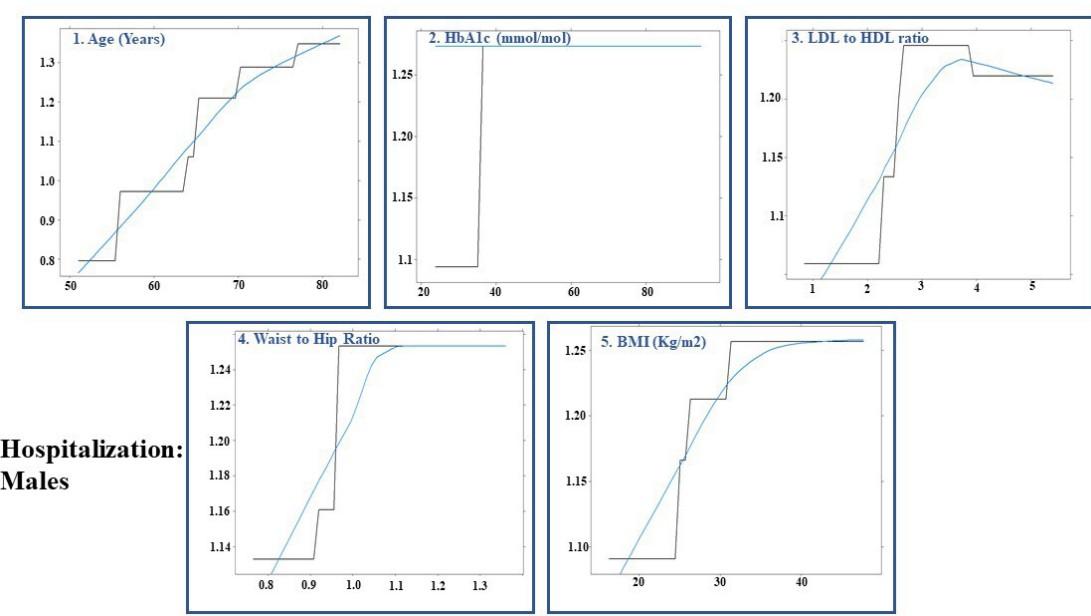

**Figure 6** Hospitalisation of males: partial dependency plot for predicting hospitalised patients in test-positive males for first five most influential variables using permutation methods using shrinkage=0.1, bag fraction=0.5, interaction depth=5, cross-validation fold=10, train fraction=0.8 and n.minobsinnode=15 as hyperparameters from grid search. BMI, body mass index; HbA1c, haemoglobin A1c; HDL, high-density lipoprotein; LDL, low-density lipoprotein.

## DISCUSSION

The present evaluation of individuals tested for SARS-CoV-2 in a UK cohort demonstrates the importance of gender-related factors along with clinical characteristics in predicting COVID-19 test positivity, hence providing guidance for identifying vaccination priority groups in the general population. While factors related to the gender role of individuals were the most influential determinants in females, cardiometabolic risk factors played a key role in males. Such a sex-specific cluster of factors associated with adverse outcomes was attenuated when considering the rate of COVID-19-related hospitalisation among test-positive individuals. Notably, older age and cardiometabolic diseases, including diabetes, obesity and dyslipidaemia, were most influential regardless of sex.

Emerging evidence has shown sex differences in contracting and severity of the infection. While most investigations have focused on biological factors as the potential culprit, few have incorporated gender determinants. Various modulating mechanisms have been suggested, including genetic factors (hormone-regulated expression of genes), the difference in innate and adaptive immune responses, as well as gendered factors such as lifestyle, behavioural and psychosocial factors.[11 13 39]

Our findings reinforce and advance the current evidence related to the impact of metabolic comorbidities and older age in the COVID-19 pandemic. Obesity has been linked with impaired pulmonary function and suppression of immune response and has been recognised as one of the most important factors in contracting COVID-19 infection, the severity of the disease and mortality.[4 40] Higher BMI and WHR are some of the more influential measures that have shown a dose–response relationship with test positivity and disease severity.[4 6 40] A study on the UKB data demonstrated a more than 50% increase in COVID-19 infection in obese and severely obese patients compared with non-obese individuals.[40] Investigation of different databases in various countries have also reported older age and comorbidities as the most important factors associated with clinical severity and mortality.[41] Specifically, diabetes has been associated with more severe disease manifestation.[42] This is consistent with our results, which demonstrated HbA1c level as one of the most prominent factors for hospitalisation in the overall and sex-stratified models. Preliminary studies have demonstrated the association between HDL level, infection and disease severity. While the mechanism of this correlation is unknown, this can be due to the antioxidant, antithrombotic and anti-inflammatory role of HDL cholesterol.[43]

A major and novel contribution of this study is the application of a sex-oriented and gender-oriented lens to inform the understanding of COVID-19 infection by conducting sex-disaggregated analyses and incorporating both sex-related and gender-related determinants. Studies have shown socioeconomic disadvantages such as living in a more deprived area and lower education to be associated with increased risk of infection and disease severity[2 4]; these factors prevail in individuals more likely to work in service-based occupations, be self-employed or live in crowded households.[3 4 44] Moreover, less access to healthcare is another factor that leads to greater infection risk and worst outcomes. By the same token, our study demonstrated that those who live in a more deprived area with lower SES and education were more susceptible to COVID-19 infection, but the impact of gender determinants was more significant among females. Women's role as caregivers within family and society increases their risk of infection.[39] Moreover, women are more likely to work as frontline workers, including nursing positions, increasing their exposure to the virus.[3 22 39] In our current study, environment score, household arrangement, working night shifts and income were among the most important factors for females. In contrast, obesity, LDL to HDL ratio and alcohol consumption were among the most influential factors in men.

The results of this study serve as an important guide for vaccination prioritisation policies. While essential workers and elderly individuals have already been targeted for vaccination, the next step will be the identification and prompt vaccination of individuals in higher risk groups in the general population. Although factors such as diabetes and obesity might be important, psychosocial risk factors such as lower SES, education level, living in a more deprived environment, risky occupation and household crowding should also be taken into account. Individuals, especially females, exhibiting such high-risk gendered factors should be prioritised for vaccination.

### Strengths and limitations

A unique feature of the study is the investigation of numerous lifestyles, socioeconomic, mental and behavioural factors representing different dimensions of gender in conjunction with clinical and laboratory factors made feasible through machine learning algorithms. Furthermore, the assessment of sex-stratified algorithms to elucidate the most influential factors in both sexes adds novel information as it has rarely been done to date.

The study should also be interpreted considering some limitations. The first limitation of this study is selection bias due to the lack of systematic and random testing across the UK. Second, this analysis was done using the baseline data that were collected from 2006 to 2010 and not the ones done at the time of COVID-19 infection diagnosis. Therefore, misclassification of determinants is possible. However, previous studies of the UKB have shown a high correlation between baseline and follow-up data for a subsample of patients who had further visits for imaging.[22] Moreover, a disproportionately higher representation of Caucasians in the study may make the results less generalisable to other ethnic groups. Finally, though we attempted to test various clinical, social and demographic factors to predict test positivity, it is essential to note that acquisition of infection is a multifactorial phenomenon that cannot be easily encoded using a small set of variables. Our relatively low predictive model

performance—as expected—reflects other influences not captured in the UKB. Similar results were obtained using the XGboost method from a recent study on the UKB dataset whereby slightly superior performance to our gradient boosting machine model was obtained, which further supports the interpretation that this is the expected accuracy for predictive models on this dataset.[18] While similar features were obtained for predicting severity (hospitalisation and fatality) in this study, combining mortality with hospitalisation for assessing severity can justify better model performance compared with only hospitalisation in our models. The difference observed in the performance of our model compared with the aforementioned study may also be explained by the lower power and heterogeneity in our sample. Moreover, since the predictive accuracy of the model is slightly low, the risk factors deduced may not be strong enough to predict the outcomes.

Additionally, with the emergence of the dominant delta and omicron variants, further studies are needed to elucidate the risk factors, though we suspect them to remain the same. Finally, the small sample size for this analysis may limit generalisability.

## CONCLUSIONS

Sex-specific risk patterns of COVID-19 test positivity exist, with gender-related factors being more relevant in females and biological factors in males. Specifically, SES, education level, number of people living in a household and high-risk jobs were associated with a higher likelihood of contracting the infection in female individuals, whereas cardiometabolic disease and obesity were more associated in males. The rate of COVID-19-related hospitalisation recognised similar favouring clinical factors regardless of sex. This study highlights the importance of prioritising high-risk groups using psychosocial determinants along with clinical factors as a targeted approach for vaccination of more at-risk population to contain the SARS-CoV-2 pandemic.

**Author affiliations**
[1]Centre for Outcomes Research and Evaluation, McGill University Health Centre, Montreal, Québec, Canada
[2]Department of Biology, McGill University, Montreal, Québec, Canada
[3]Faculty of Medicine and Health Sciences, McGill University, Montreal, Québec, Canada
[4]Augmented Intelligence & Precision Health Laboratory (AIPHL), Department of Radiology, McGill University Health Centre, Montreal, Québec, Canada
[5]Department of Translational Medicine, University of Ferrara, Ferrara, Italy
[6]Faculty of Nursing, University of Alberta, Edmonton, Alberta, Canada
[7]Heart and Stroke Strategic Clinical Networks, Alberta Health Services, Edmonton, Alberta, Canada
[8]Divisions of Clinical Epidemiology and General Internal Medicine, McGill University Health Centre Research Institute, Montreal, Quebec, Canada
[9]Electronic Health Information Laboratory, Children's Hospital of Eastern Ontario Research Institute, Ottawa, Ontario, Canada
[10]School of Epidemiology and Public Health, University of Ottawa, Ottawa, Ontario, Canada

**Collaborators** The GOING-FWD investigators: Alexandra Kautzky-Willer, Medical University of Vienna, Austria; Karolina Kublickiene, Karolinska Institutet, Sweden; Maria Trinidad Herrero, Universidad de Murcia, Spain.

**Contributors** ZA, YS, PA, CN, VR, LP, KEE and GOING-FWD investigators designed the study, developed the questions and contributed to the development of conceptual framework and writing the first draft. ZA, YS and KEE performed the main analysis and critical revisions of the manuscript. FM and RF validated the results by performing secondary analyses and finalising the draft. ZA, YS, PA, CN, VR, LP, KEE and GOING-FWD investigators supervised the project from study design to final draft and approved the final version of the manuscript. The KEE and LP are the guarantor of the study and accept full responsibility for the work and/or the conduct of the study, had access to the data, and controlled the decision to publish. All authors agree to be accountable for all aspects of the work by ensuring that questions related to the accuracy or integrity of any part of the work are appropriately investigated and resolved.

**Funding** The GOING-FWD Consortium is funded by the GENDER-NET Plus ERA-NET Initiative (project ref. number: GNP-78), the Canadian Institutes of Health Research (GNP-161904), 'La Caixa' Foundation (ID 100010434) with code LCF/PR/DE18/52010001, the Swedish Research Council (2018-00932) and the Austrian Science Fund (FWF, I 4209). VR was funded by the Scientific Independence of Young Researcher Program of the Italian Ministry of University, Education, and Research (RBSI14HNVT).

**Competing interests** None declared.

**Patient and public involvement** Patients and/or the public were not involved in the design, or conduct, or reporting, or dissemination plans of this research.

**Patient consent for publication** Not required.

**Ethics approval** This study involves human participants and ethics approval for the project was obtained from the coordinator centre at McGill University, Canada (REB # 2020-5452 (Gender Outcomes INternational Group: to Further Well-being Development (GOING-FWD), CIHR GNP-161904, 01 October 2018–31 March 2023). The data were obtained from the UK Biobank repository (application ID: 45551). Participants gave informed consent to participate in the study before taking part.

**Provenance and peer review** Not commissioned; externally peer reviewed.

**Data availability statement** No data are available.

**ORCID iDs**
Louise Pilote http://orcid.org/0000-0002-6159-0628
Khaled El Emam http://orcid.org/0000-0003-3325-4149

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
