## [Reviewer comments · BMJ Open]

ARTICLE DETAILS

TITLE (PROVISIONAL)	The Importance of Sex and Gender Factors for COVID-19 Infection and Hospitalization: A Sex-stratified Analysis Using Machine Learning in UK Biobank data
AUTHORS	Azizi, Zahra; Shiba, Yumika; Alipour, Pouria; Maleki, Farhad; Raparelli, Valeria; Norris, Colleen; Forghani, Reza; Pilote, Louise; El Emam, Khaled; Investigators, GOING FWD

VERSION 1 – REVIEW

REVIEWER	Bentham, James University of Kent , School of Mathematics, Statistics and Actuarial Science
REVIEW RETURNED	19-Jul-2021

GENERAL COMMENTS	The authors present an interesting study of factors affecting Covid vaccination. I have the following comments: 1. The text should be checked thoroughly for spelling and grammatical mistakes, e.g., in the abstract, results "29.4% were test positive".2. The conclusion in the abstract is slightly confusing at present. It should be made clearer that gender-related variables affect females in particular.3. The authors should describe any corrections of p-values for multiple testing, or justify why this wasn't carried out.4. Machine learning methods have been used, but are these any more effective than simpler methods such as logistic regression? This should be discussed.5. In results, p9, it's not clear why the age has a confidence interval (68.5+/-8.88). This should be clarified.6. In Table 1, I think the overall number testing positive is 3184 (not 3148)7. The numbers of decimal places should be consistent in the tables8. References should be checked to ensure they have full details9. Further description of the figures should be provided, as it's not clear how to interpret them.10. Figure 3 has ethnicity described as African-American, but this is UK data
--

REVIEWER	So, Hon-Cheong The Chinese University of Hong Kong
REVIEW RETURNED	17-Oct-2021

GENERAL COMMENTS	Although there are quite a lot of similar research including sex or gender as risk factor to build machine learning risk predictive model for COVID-19, there are few research papers that focus on gender-
---

stratified analysis of the risk factors. It is one of the main novelties of this paper, and the paper is generally well-written. However, the relatively small sample size (and no. of cases) may hinder the generalization of the research findings, power to detect risk factors and predictive accuracy, which is a limitation that should be mentioned too.

Here are more detailed comments/suggestions:

- Page 3, Sentence “previous studies of the UK Biobank have shown a high correlation ~”)
 - o Please cite the studies.
- Page 3, Sentence “our relatively lower predictive model performance ~”)
 - o It is better to mention the AUC in the main text as well.
- Page 3, Sentence “A recent study in UK Biobank revealed similar performance in the gbm model ~”)
 - o Please cite the study
- Page 4, Sentence “The novel coronavirus SARS-CoV-2 (COVID-19) pandemic has led to more than 105 million reported~”)
 - o Please mention the cut-off date
- Page 7, Sentence “Baseline comorbidities were self-reported and collected using a touch screen device~”)
 - o Advise to include references or literature support on why such comorbid traits were selected
- Page 7, Sentence “Case analysis (pairwise deletion) approach was used for dealing with missing data~”)
 - o Pairwise deletion can be applied in student t-test, but is it also used for building ML model? This may lead to a large no. of observations deleted. Could you state how many observations were deleted and describe why this approach was used (instead of imputation)?
- Page 8, Sentence “To alleviate the effect of class imbalance on the model development 28, an oversampling approach was used~”)
 - o Please mention which oversampling approach is used (e.g. Random, bootstrapping, Smote, KMeans Smote ...etc) and the details.
- Page 8, Sentence “We also used a grid-search procedure to find the best combination of boosting hyperparameters and tree specific hyper-parameters~”)
 - o Could you clarify what dataset used for hyper-parameter tuning? If the testing set is involved, the hyperparameters will be overfitted to the testing sets. Also, please mention the grid-search parameters.
- Page 8, Sentence “derived metrics including accuracy, precision, recall (sensitivity), specificity as well as AUC score~”)
 - o In the model for predicting hospitalization, the case:control ratio is ~7:3 which is bit imbalanced. May also include AUPRC, and what cutoff of the predicted probability is used to determine the precision/recall, sensitivity/specificity?
- Page 9, Sentence “In descriptive analyses, amongst all patients, those older than 80 years, minority ethnicities, and individuals born outside of the UK had a higher infection rate~”)
 - o Please state and explain the descriptive statistic that supports the findings, such as “those older than 80 years, minority ethnicities ... had a higher infection rate”.
- Page 9, Sentence “The performance of the gradient boosting machine models is summarized in Table 3.~”)
 - o Interpretation of the results in Table 3 in the main text?
- Page 9, Sentence “The prediction models for the overall study population suggest that~”)
 - o To enhance the readability, Figure 1 should be mentioned before

	interpreting the results.  • Page 10, Sentence “The performance of the gradient boosting machine models is summarized in Table 4.~” o Interpretation of the results in Table 4 in the main text? • Page 13, Section “Strengths and Limitations” o Relatively small number of samples (n=4510) and cases used to build the predictive model and the relatively low prediction accuracy should be mentioned in this section. Both issues will affect the prediction accuracy when applying the model to the population for prioritizing vaccination policy. o due to the low prediction accuracy of the model, the risk factors deduced from the VarImp or Partial dependency plot in figure 1-6 may not be reliable and robust enough o the delta variant is predominant now, while we expect similar risk factors, this require further studies • Page 14, Sentence “supports the interpretation that this is the expected accuracy for predictive models on this Dataset18 ~” o Referring to these papers (https://doi.org/10.1016/j.imu.2021.100564 & https://doi.org/10.1136/bmj.m1328), the AUCs in these previous works are higher than those results specified in Table 3 and 4 (i.e., AUC ranged from 0.563 to 0.603 for the CV models). o In the cited paper [18], the AUCs of the CV models are ranged from 0.696 to 0.818 which are much higher than the best performing model in your study (i.e., AUC ranged from 0.563 to 0.603 for the CV models). So, this paper [18] may not support the interpretation that this is the expected accuracy for predictive models on this dataset. Please kindly provide a more detailed explanation, eg heterogeneity in the sample or other characteristics For the PDP plots Resolution appears to be too low. What's on the y-axis and the unit (and how could we interpret it)? What's the blue curve (?best-fitted curve by whichever method) Note: this review is completed by me together with Mr. Kenneth C.Y. Wong
--	---

VERSION 1 – AUTHOR RESPONSE

Response to Reviewer: 1

Dr. James Bentham, University of Kent

Comments to the Author:

The authors present an interesting study of factors affecting Covid vaccination. I have the following comments:

1. The text should be checked thoroughly for spelling and grammatical mistakes, e.g., in the abstract, results "29.4% were test positive".

Reply: Thank you for your comment that gives us the opportunity to clarify the added value of the manuscript. We have reviewed the manuscript and revised spelling and grammatical errors.

2. The conclusion in the abstract is slightly confusing at present. It should be made clearer that gender-related variables affect females in particular.

Reply: The following changes were made in the text.

“Gender-related factors have a greater impact in females; hence they should be considered in identifying priority groups for COVID-19 infection vaccination campaigns.”

3. The authors should describe any corrections of p-values for multiple testing or justify why this wasn't carried out.

Reply: We adjusted the p-values in Table 1 and 2 based on Bonferroni type adjustments to correct for multiple testing and the significant results in the table are shown with * to signify adjustment. The statement below was also added to the methods.

“Bonferroni type adjustments were utilized to correct for multiple testing and the results after adjustment were used for interpretation.”

Table1: *Bonferroni Correction= $0.05/24$ (number of variables) =0.002. Based on this correction the differences marked with * are considered statistically significant.

Table2: *Bonferroni Correction= $0.05/18$ (number of variables) =0.003. Based on this correction the differences marked with * are considered statistically significant.

4. Machine learning methods have been used, but are these any more effective than simpler methods such as logistic regression? This should be discussed.

Reply: In contrast to logistic regression, machine learning models apply more sophisticated algorithms which take into consideration interconnectivity of various risk factors and comorbidities. This was especially important in Covid infection- related risk factors as they are highly influenced by various environmental and physiological factors.

5. In results, p9, it's not clear why the age has a confidence interval (68.5+/-8.88). This should be clarified.

Reply: To clarify, this is standard deviation not CI. We added a statement specifying this for Table1,2,3,4

Table 1,2: Results are presented as mean \pm standard deviation for numerical variables and percentage for categorical variables.

Table 3,4: *Area Under Curve (AUC) is reported with 95% Confidence Interval (CI)

6. In Table 1, I think the overall number testing positive is 3184 (not 3148)

Reply: Thank you. We have made the correction.

7. The numbers of decimal places should be consistent in the tables

Reply: We made the changes accordingly in Table 1, and 2.

8. References should be checked to ensure they have full details

Reply: We checked all references and updated the details.

)

9. Further description of the figures should be provided, as it's not clear how to interpret them.

Reply: A more detailed description is added to the methods.

“The Y axis shows how the predicted value changes with change in predictor variables. If the line in the plot is constant near zero it means that the variable has no effect on the model. A negative value means that a specific value of the predictor variable is less likely to predict the correct class of outcome, whereas a positive value means the predictor variable has a positive impact on predicting the correct class (36). A LOESS (locally estimated scatterplot smoothing) line is fit to show the trend.”

10. Figure 3 has ethnicity described as African-American, but this is UK data

Reply: Changes were made to reflect the ethnicity of British population. We replaced the African American to Black British.

Reviewer: 2

Dr. Hon-Cheong So, The Chinese University of Hong Kong

Comments to the Author:

Although there are quite a lot of similar research including sex or gender as risk factor to build machine learning risk predictive model for COVID-19, there are few research papers that focus on gender-stratified analysis of the risk factors. It is one of the main novelties of this paper, and the paper is generally well-written. However, the relatively small sample size (and no. of cases) may hinder the generalization of the research findings, power to detect risk factors and predictive accuracy, which is a limitation that should be mentioned too.

Here are more detailed comments/suggestions:

- Page 3, Sentence “previous studies of the UK Biobank have shown a high correlation ~”)
 - o Please cite the studies.

Reply: Thank you very much for your comment. Page 3 is the summary of strength and limitations. The reference for the sentence is in the limitation part of discussion as below.

“The study should also be interpreted in light of some limitations. The first limitation of this study is selection bias due to the lack of systematic and random testing across the UK. Second, this analysis was done using the baseline data that was collected from 2006 to 2010 and not the ones done at the time of COVID infection diagnosis. Therefore, misclassification of determinants is possible. However, previous studies of the UK Biobank have shown a high correlation between baseline and follow-up data for a subsample of patients who had further visits for imaging 22 Finally , the small sample size for this analysis may limit generalizability.

22. Mutambudzi M, Niedzwiedz CL, Macdonald EB, et al. Occupation and risk of COVID-19: prospective cohort study of 120,621 UK Biobank participants. medRxiv 2020.”

- Page 3, Sentence “our relatively lower predictive model performance ~”)
 - o It is better to mention the AUC in the main text as well.

Reply: We provided the AUC on Page 3 as requested.

“Finally, our relatively lower predictive model performance (Test positivity: AUC: 0.570 (95% Confidence Interval (CI): 0.537-0.604), Hospitalization: AUC: 0.60 (95% CI: 0.534-0.665))—as expected—reflects other influences not captured in the UK Biobank. However, a recent study in a larger UK Biobank sample size revealed enhanced performance of the same type of machine learning model which illustrates power limitation in an otherwise accurate model.”

- Page 3, Sentence “A recent study in UK Biobank revealed similar performance in the gbm model ~”)
 - o Please cite the study

Reply:.. This is the summary of the strength and weakness; more detailed explanation has been added to the discussion. We removed this part from summary section.

“Our relatively low predictive model performance—as expected—reflects other influences not captured in the UK Biobank. Similar results were obtained using the XGboost method from a recent study on the UK Biobank dataset whereby slightly superior performance to our gbm model was obtained, which further supports the interpretation that this is the expected accuracy for predictive models on this dataset¹⁸. While, similar features were obtained for predicting severity (hospitalization and fatality) in this study, combining mortality with hospitalization for assessing severity can justify better model performance compared to only hospitalization in our models.”

- Page 4, Sentence “The novel coronavirus SARS-CoV-2 (COVID-19) pandemic has led to more than 105 million reported~”)
 - o Please mention the cut-off date

Reply: We added the new date and cut-off date.

“more than 250 million reported positive cases and over five million deaths worldwide as of November

2021”

- Page 7, Sentence “Baseline comorbidities were self-reported and collected using a touch screen device~”)

- o Advise to include references or literature support on why such comorbid traits were selected
Reply: We added the following statement to support our choice of selected comorbidities and variables.

“These variables were selected based on their significance as demonstrated by a number of previous investigations on the UK Biobank 2 3 5-8 18 22 25-27.”

- Page 7, Sentence “Case analysis (pairwise deletion) approach was used for dealing with missing data~”)

- o Pairwise deletion can be applied in student t-test, but is it also used for building ML model? This may lead to a large no. of observations deleted. Could you state how many observations were deleted and describe why this approach was used (instead of imputation)?

Reply: Thank you very much for your comment. We solely used this method for our T-test comparisons. As for gbm model, it deals with missing data automatically during the tree development, and therefore there is no need to impute them. Each split has three nodes, left, right, and missing. Therefore, missing values are treated as a separate value and modeled. See and the link to GBM documentation.

<https://stats.stackexchange.com/questions/322585/r-gbm-handling-of-missing-values>

- Page 8, Sentence “To alleviate the effect of class imbalance on the model development 28, an oversampling approach was used~”)

- o Please mention which oversampling approach is used (e.g. Random, bootstrapping, Smote, KMeans Smote ...etc) and the details.

Reply: The bootstrapping method was used for oversampling approach. The following statement was added.

“To reduce the effect of class imbalance on model development 31, a bootstrap oversampling approach was used. For each endpoint, i.e. test-positivity or hospitalization, three different models were developed: (1) a model for males, (2) a model for females, and (3) an overall model for males and females combined. Calibration was then performed using isotonic regression method.”

- Page 8, Sentence “We also used a grid-search procedure to find the best combination of boosting hyperparameters and tree specific hyper-parameters~”)

- o Could you clarify what dataset used for hyper-parameter tuning? If the testing set is involved, the hyperparameters will be overfitted to the testing sets. Also, please mention the grid-search parameters.

Reply: We used training set with cross validation for our grid search. The following parameters were used for our grid search.

```
hyper_grid <- expand.grid(  
  shrinkage = c(0.3, 0.1, 0.05, 0.01, 0.001),  
  interaction.depth = c(3, 5, 7),  
  n.minobsinnode = c(5, 10, 15),  
  bag.fraction = c(.65, 0.5, .8, 1),  
  trees = 0,  
  time = 0,  
  optimal_trees=0,  
  roc=0,  
  logloss=0  
)
```

- Page 8, Sentence “derived metrics including accuracy, precision, recall (sensitivity), specificity as well as AUC score~”)

- o In the model for predicting hospitalization, the case:control ratio is ~7:3 which is bit imbalanced. May

also include AUPRC, and what cutoff of the predicted probability is used to determine the precision/recall, sensitivity/specificity?

Reply: We used the best threshold of ROC as cut off for determining precision/recall, sensitivity/specificity (Table 3,4). Another metric which focuses on prediction of the positive class is the Area Under the Precision-Recall Curve (AUPRC) [1]. The interpretation of AUPRC is dependent on the class distribution of the outcome as the minimal achievable value is dependent on that distribution [2], and the AUPRC value of a random classifier is the rate of the positive class [2]

References

[1] Davis J, Goadrich M, "The relationship between Precision-Recall and ROC curves," in Proceedings of the 23rd international conference on Machine learning, New York, NY, USA, Jun. 2006, pp. 233–240. doi: 10.1145/1143844.1143874.

[2] Boyd K, Santos Costa V, Davis J, and Page C. D. , "Unachievable Region in Precision-Recall Space and Its Effect on Empirical Evaluation," Proc Int Conf Mach Learn, vol. 2012, p. 349, Dec. 2012.

"We used the best threshold of ROC as cut off for determining precision/recall, sensitivity/specificity. Another metric which focuses on predictions of the positive class is the Area Under the Precision-Recall Curve (AUPRC) 34. Interpretation of AUPRC is dependent on the class distribution of the outcome as the minimal achievable value is dependent on that distribution 35, and the AUPRC value of a random classifier is the rate of the positive class"

- Page 9, Sentence "In descriptive analyses, amongst all patients, those older than 80 years, minority ethnicities, and individuals born outside of the UK had a higher infection rate~")

- o Please state and explain the descriptive statistic that supports the findings, such as "those older than 80 years, minority ethnicities ... had a higher infection rate".

Reply: Thank you very much for your comment.

"In descriptive analyses, there was a difference in age between test-positive and test-negative individuals ($p < 0.001$), and specifically those younger than 60 years (Test positive vs negative: 28% vs 21.8%) and those older than 80 years (Test + vs -: 6.2% vs 5.7%). Similarly there was significant difference in test positivity amongst ethnicities (minority ethnicity: Test + vs -: 13.2% vs 7.6% $p < 0.001$), and individuals born outside of the UK (Test + vs -: 14.3% vs 9.8%, $p < 0.001$). Moreover, infected patients had a higher BMI (Test + vs -: 28.9 ± 5.4 vs 28.2 ± 5.4 , $p < 0.001$), WHR (Test + vs -: 0.89 ± 0.09 vs 0.88 ± 0.09), HbA1C (Test + vs -: 37.8 ± 9.1 vs 37.1 ± 8.3 , $p = 0.03$), LDL to HDL ratio (Test + vs -: 2.7 ± 0.8 vs 2.5 ± 0.9 , $p < 0.001$), and lower HDL (Test + vs -: 1.3 ± 0.3 vs 1.4 ± 0.3 , $p < 0.001$) levels. On the other hand, smokers (Test + vs -: 11% vs 14%, $P = 0.02$), alcohol drinkers (Test + vs -: 15.8% vs 20.8%, $p < 0.001$), more educated individuals (Test + vs -: 31.2% vs 35.9%, $p < 0.001$), house owners (Test + vs -: 79.6% vs 82.2%, $p = 0.04$), and those who did not participate in any social activity (Test + vs -: 31.8% vs 32.7%, $p = 0.003$) had a lower rate of infection. Test positive patients tended to have a greater number of people in their household (Test + vs -: 2.5 ± 1.3 vs 2.3 ± 1.5 , $p < 0.001$) and live in more deprived areas ($p < 0.001$)."

- Page 9, Sentence "The performance of the gradient boosting machine models is summarized in Table 3.~")

- o Interpretation of the results in Table 3 in the main text?

Reply: The following statement was added to the text.

"The area under curve (AUC) for test positivity in the overall model, male and female specific models were 0.570 (95% Confidence Interval (CI): 0.537-0.604), 0.575 (0.529-0.621), 0.561 (95% CI: 0.512-0.609), respectively."

- Page 9, Sentence "The prediction models for the overall study population suggest that~")

o To enhance the readability, Figure 1 should be mentioned before interpreting the results.

Reply: The following statement was changed in the text.

“Figure 1 illustrates the partial dependence plots used for interpreting the results and direction of each variable in the models.”

• Page 10, Sentence “The performance of the gradient boosting machine models is summarized in Table 4.”

o Interpretation of the results in Table 4 in the main text?

Reply: Thank you very much for your comment. The following statement was added to the text.

“The area under curve (AUC) for hospitalization in test positive patients in overall, male, and female specific models were 0.60 (95% CI: 0.534-0.665), 0.544 (95% CI: 0.453-0.635), and 0.612 (95% CI: 0.532-0.692), respectively.”

• Page 13, Section “Strengths and Limitations”

o Relatively small number of samples (n=4510) and cases used to build the predictive model and the relatively low prediction accuracy should be mentioned in this section. Both issues will affect the prediction accuracy when applying the model to the population for prioritizing vaccination policy.

o due to the low prediction accuracy of the model, the risk factors deduced from the VarImp or Partial dependency plot in figure 1-6 may not be reliable and robust enough

o the delta variant is predominant now, while we expect similar risk factors, this require further studies

Reply: The following statement was added to the text.

“The difference observed in the performance of our model compared to the aforementioned study maybe also explained by the lower power and heterogeneity in our sample. Moreover, since the predictive accuracy of the model is slightly low, the risk factors deduced may not be strong enough to predict the outcomes. Additionally with the emergence of the dominant delta variant further studies are needed to elucidate the risk factors, though we suspect them to remain the same. Finally, the small sample size for this analysis may limit generalizability”

• Page 14, Sentence “supports the interpretation that this is the expected accuracy for predictive models on this Dataset18 ~”

o Referring to these papers (<https://doi.org/10.1016/j.imu.2021.100564> & <https://doi.org/10.1136/bmj.m1328>), the AUCs in these previous works are higher than those results specified in Table 3 and 4 (i.e., AUC ranged from 0.563 to 0.603 for the CV models).

o In the cited paper [18], the AUCs of the CV models are ranged from 0.696 to 0.818 which are much higher than the best performing model in your study (i.e., AUC ranged from 0.563 to 0.603 for the CV models). So, this paper [18] may not support the interpretation that this is the expected accuracy for predictive models on this dataset. Please kindly provide a more detailed explanation, eg heterogeneity in the sample or other characteristics

Reply: Thank you very much for your comment. The following changes were made to address the issue.

“Our relatively low predictive model performance—as expected—reflects other influences not captured in the UK Biobank. Similar results were obtained using the XGboost method from a recent study on the UK Biobank dataset whereby slightly superior performance to our gbm model was obtained, which further supports the interpretation that this is the expected accuracy for predictive models on this dataset18. While, similar features were obtained for predicting severity (hospitalization and fatality) in this study, combining mortality with hospitalization for assessing severity can justify better model performance compared to only hospitalization in our models. The difference observed in the performance of our model compared to the aforementioned study maybe also explained by the lower power and heterogeneity in our sample. Moreover, since the predictive accuracy of the model is slightly low, the risk factors deduced may not be strong enough to predict the outcomes. Additionally with the emergence of the dominant delta variant further studies are needed to elucidate the risk factors, though we suspect them to remain the same. Finally, the small sample size for this analysis may limit generalizability.”

For the PDP plots

Resolution appears to be too low. What's on the y-axis and the unit (and how could we interpret it)?
What's the blue curve (?best-fitted curve by whichever method)

Reply: The following statement was added to figure legends and methods to address the issue. We also increased the pixels per inch for the images and hence the resolution should now be enhanced.

“The Y axis shows how the predicted value changes with change in predictor variables. If the line in the plot is constant near zero it means that the variable has no effect on the model. A negative value means that a specific value of the predictor variable is less likely to predict the correct class of outcome, whereas a positive value means the predictor variable has a positive impact on predicting the correct class (36.) A LOESS (locally estimated scatterplot smoothing) line is fit to show the trend. The number beside each variable shows the order of feature importance and most influential variables for each model. For identifying the most influential variables, permutation-based feature importance was utilized. This approach measures a feature (variable) importance by calculating the increase of the model’s prediction error after permuting the feature.”

Note: this review is completed by me together with Mr. Kenneth C.Y. Wong

VERSION 2 – REVIEW

REVIEWER	Bentham, James University of Kent , School of Mathematics, Statistics and Actuarial Science
REVIEW RETURNED	03-Jan-2022
GENERAL COMMENTS	The authors have answered all of my comments and the paper is ready for publication. Two minor comments: the paper should be proof-read thoroughly as there are still some minor grammatical mistakes (e.g. World Health Organization in lower case). Also, the delta variant is mentioned in the discussion, so this could be updated to include omicron
REVIEWER	So, Hon-Cheong The Chinese University of Hong Kong
REVIEW RETURNED	03-Jan-2022
GENERAL COMMENTS	We thank the authors for their careful consideration of our comments and thorough revisions. Most of our comments are addressed adequately. One minor point on this comment: Page 8, Sentence “We also used a grid-search procedure to find the best combination of boosting hyperparameters and tree specific hyper-parameters~” Is a validation dataset that does not belonging to the training and testing dataset used for hyper-parameter tuning? Otherwise, the hyperparameters will be overfitted to the testing sets. > Maybe I missed it, but I suggest the authors mentioned the grid

	search parameters also in the main text or supp info. Also, please mention that the dataset is split into training and testing set (only), and the best model was chosen from test set performance. Ideally, a separate 'tuning' set should be set aside for tuning hyper-parameters to avoid optimistic bias, but if this is not done, I appreciate if the authors could mention this as a limitation or future work.
--	--

VERSION 2 – AUTHOR RESPONSE

Reviewer: 2

Dr. Hon-Cheong So, The Chinese University of Hong Kong

Comments to the Author:

We thank the authors for their careful consideration of our comments and thorough revisions. Most of our comments are addressed adequately. One minor point on this comment:

Page 8, Sentence “We also used a grid-search procedure to find the best combination of boosting hyperparameters and tree specific hyper-parameters~”)

Is a validation dataset that does not belonging to the training and testing dataset used for hyper-parameter tuning? Otherwise, the hyperparameters will be overfitted to the testing sets.

> Maybe I missed it, but I suggest the authors mentioned the grid search parameters also in the main text or supp info. Also, please mention that the dataset is split into training and testing set (only), and the best model was chosen from test set performance. Ideally, a separate 'tuning' set should be set aside for tuning hyper-parameters to avoid optimistic bias, but if this is not done, I appreciate if the authors could mention this as a limitation or future work.

Reply: Thank you for your comment that gives us the opportunity to clarify the added value of the manuscript. We have used a 5-fold cross-validation using the training dataset for hyper-parameter tuning using a grid search. The data was indeed split to 70% train and 30% test.

Page8: “The data was split into 70% training and 30% test sets, where the test set was only used for the evaluation of the final models. We also used a grid-search procedure to find the best combination hyper-parameters (e.g., learning rate, interaction depth, bagging fraction, and the minimum number of observations in terminal nodes) (supplement2) using 5-fold cross-validation on the training dataset and the AUROC metric as the criterion.”

Supplement 2: Grid Search Parameters

Shrinkage 0.3 0.1 0.05 0.01 0.001

Interaction Depth 3 5 7

N.minobsinnode 5 10 15

Bag.fraction 0.65 0.5 0.8 1

Reviewer: 1

Dr. James Bentham, University of Kent

Comments to the Author:

The authors have answered all of my comments and the paper is ready for publication.

Two minor comments: the paper should be proof-read thoroughly as there are still some minor grammatical mistakes (e.g. World Health Organization in lower case). Also, the delta variant is mentioned in the discussion, so this could be updated to include omicron.

Reply: Thank you for your comment that gives us the opportunity to clarify the added value of the manuscript. We have reviewed the manuscript and revised spelling and grammatical errors. We also added the omicron variant as the current dominant variant.

Page 15: "Additionally with the emergence of the dominant delta and omicron variant further studies are needed to elucidate the risk factors, though we suspect them to remain the same."